# Taper Function for *Pinus nigra* in Central Italy: Is a More Complex Computational System Required?

**Maurizio Marchi** [1],*, **Roberto Scotti** [2], **Giulia Rinaldini** [3] and **Paolo Cantiani** [4]

[1] CNR - Institute of Biosciences and BioResources (IBBR), Florence division, Via Madonna del Piano 10, I-50019 Sesto Fiorentino (Firenze), Italy

[2] Nuoro Forestry School, Department of Agriculture, University of Sassari, Via C. Colombo 1, I-08100 Nuoro, Italy; scotti@uniss.it

[3] Department of Agriculture, Food, Environment and Forestry (DAGRI), University of Florence, Via San Bonaventura 13, I-50145 Firenze, Italy; giulia.rinaldini@unifi.it

[4] CREA - Research Centre for Forestry and Wood, Viale S. Margherita 80, I-52100 Arezzo, Italy; paolo.cantiani@crea.gov.it

* Correspondence: maurizio.marchi@cnr.it; Tel.: +39-055-522-5746

**Abstract:** Stem tapers are mathematical functions modelling the relative decrease of diameter (rD) as the relative height (rH) increase in trees and can be successfully used in precision forest harvesting. In this paper, the diameters of the stem at various height of 202 *Pinus nigra* trees were fully measured by means of an optical relascope (CRITERION RD 1000) by adopting a two-steps non-destructive strategy. Data were modelled with four equations including a linear model, two polynomial functions (second and third order) and the Generalised Additive Model. Predictions were also compared with the output from the TapeR R package, an object-oriented tool implementing the $\beta$-Spline functions and widely used in the literature and scientific research. Overall, the high quality of the database was detected as the most important driver for modelling with algorithms almost equivalent each other. The use of a non-destructive sampling method allowed the full measurement of all the trees necessary to build a mathematical function properly. The results clearly highlight the ability of all the tested models to reach a high statistical significance with an adjusted-R squared higher than 0.9. A very low mean relative absolute error was also calculated with a cross validation procedure and small standard deviation were associated. Substantial differences were detected with the TapeR prediction. Indeed, the use of mixed models improved the data handling with outputs not affected by autocorrelation which is one of the main issues when measuring trees profile. The profile data violate one of the basic assumptions of modelling: the independence of sampled units (i.e., autocorrelation of measured values across the stem of a tree). Consequently, the use of simple parametric equations can only be a temporary resource before more complex built-in apps are able to allow basic users to exploit more powerful modelling techniques.

**Keywords:** silviculture; ecological modelling; ecological mathematics; precision forestry; statistical sampling; optical relascopy

## 1. Introduction

The European black pine (*Pinus nigra* J.F. Arnold) sometimes reported as the Austrian pine or simply the black pine, is a long-lived tree botanically recognized as a collective species occurring across the northern part of the Mediterranean basin, from Spain to Turkey [1]. Thanks to its ability to grow on poor and bare soils, this tree has been widely used for ecological restoration activities across the whole of Europe since the beginning of the 20th century and especially after the first and second world

wars [2,3]. The main aim of many reforestation programs has been to avoid landslides and to fight land abandonment when people were moving from the countryside to the cities in the early 1950s [4]. As expected, the use of this species improved the soil quality after decades of agricultural exploitation, allowing the ecological succession and the restoration of native hardwood forest tree species of the climatic envelope, generally composed by trees belonging to the *Quercus* genus [5].

According to the data delivered with the last available national forest inventory (INFC2005), pure European black pine stands in Italy cover an area of 444,785 ha (Figure 1) representing 4.25% of the whole forest area [6], with an age ranging from 50 to 95 years. Most of these stands were established under the main reforestation program that occurred after the Second World War.

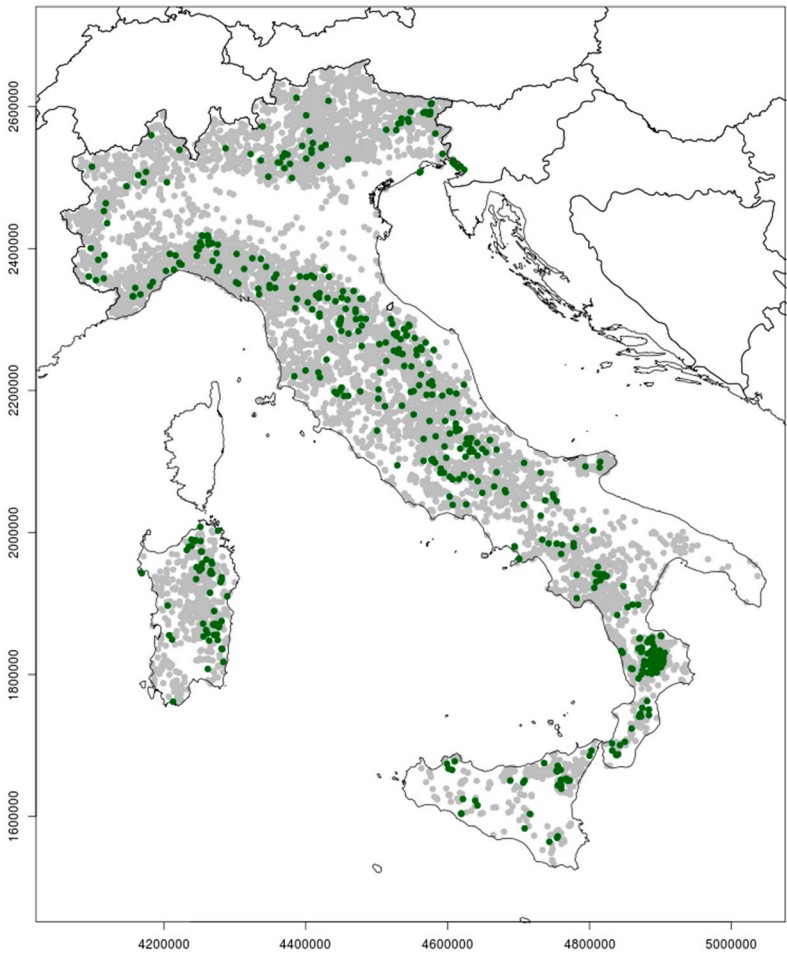

**Figure 1.** Spatial distribution of all INFC2005 inventory points (all dots) and sampling units which were classified as "Artificial black pine stands" (dark dots) during field activities.

Pandering to the well known ecological requirements of the species due to its ability to adapt to many different environments [7], seeds came from many areas of the natural range to be used in many different ecological conditions. The Calabrian provenance (*Pinus nigra* subsp. *laricio*) was collected from Southern stands populations [8] and a different ecotype was selected from Northern populations, the Austrian pine (*Pinus nigra* subsp. *nigra*), naturally occurring on the borders between Italy and Austria [9]. While the first was planted on acid soils, the second one was mainly used on calcareous conditions [3,10]. In addition, a third and ecologically intermediate variety was often used in central Italy, coming from a small and isolated population close to the town of Villetta Barrea, province of Aquila (*Pinus nigra* spp. *nigra* var. *italica*) and currently acknowledged as a marginal forest population [11]. Overall, artificial stands were established for hydrological purposes, soil protection and for social wellbeing. Unfortunately, forest management plans were rarely applied, few thinning

interventions were made in both private or public areas and consequently, the economic interest in this species decreased gradually.

Optimal assortment allocation is a key element in the wood products supply chain [12], particularly artificial stands. The use of spatial indices and models to describe the structure of a forest and to support forest management trajectories [13–16] are nowadays acknowledged as compulsory in a precision forestry framework [17,18] but research in this area, in Italy, is in serious delay [19]. Recent studies also raised the interest in the ability of *Pinus nigra* spp. to provide a wide range of ecosystem services [20,21] creating an ecologically dynamic system where biodiversity level increased quickly, thanks to all the ecological processes restored due to the artificial stands [22]. A new potential roundwood market has also been studied in addition to woodchips [5,23] and non-woody products such as truffles [24,25]. Research on modelling tools targeted at a simple and cost-effective management of European black pine trees is aimed at favouring this market.

Stem taper functions [12,19,26,27] are needed to support the decision-making process where trees are characterized according to the potential assortments to be retrieved from harvesting. Such functions are used to model the diameter reduction of the stem as distance from the ground (i.e., height along the tree) increases. Modelling stem taper and volume is crucial in many forest management and planning systems dealing with economic aspects. Taper models are used to predict the diameter size at any location along the stem. These predictions provide flexible means to estimate the volume of the stem and of any assortments potentially achievable in forest harvesting [27]. A great variety of equation structures have been proposed as the mathematical core of taper functions, estimating (for a given species, in given growing conditions) stem diameter (d(h)) at height (h) along the stem with diameter at breast height and tree height as additional input parameters. In the past, data came from felled trees where all the stem was easily accessible on the ground and measurements with a calliper or a tape were possible. Researchers were also forced to follow enterprises activities, with safety risks. Moreover, felled trees were sometimes damaged, malformed and small, such as those deriving from thinning with a potential impact on the predictive model. Nowadays, an optical tool simplified the work [12]. The data required for model calibration can be easily measured on standing trees by means of optical instruments such as Terrestrial Laser Scanning (TLS) or older instruments as Bitterlich's relascope, the Finnish parabolic calliper or optical telescopes.

This paper reports the results from a study concerning sample size optimization and taper function model complexity evaluation. In this paper, a comparative test between modelling methods was performed, running on a two-stages-derived dataset with measurement coming from a survey campaign in 2018. An optical relascope was used here (CRITERION RD-1000), allowing the sampling of 202 European black pine trees without felling. The main aim of this paper was to demonstrate the importance of an adequate sampling criteria and the low impact of the tool used to build the model (i.e., the mathematical function).

## 2. Materials and Methods

### 2.1. Sampling Method

Sample size heavily conditions research cost (or is conditioned by research funds) and in this work, specific efforts were made to use the minimum sampling size required to produce a useful result. In random sampling with uniform probabilities, the minimum number of samples required to provide an estimate of the population mean with an error not exceeding $\varepsilon$ at a confidence level of $p$ can be estimated by the following formula, also used in forest monitoring [28]:

$$n = \left(\frac{t \cdot CV}{\varepsilon}\right)^2 \tag{1}$$

where $n$ is the number of samples to be effectively measured, CV is the coefficient of variation (CV) of the parameter to be estimated, $\varepsilon$ is the relative error (= 5% in this case) and $t$ is the value of Student's

t for a given confidence level (generally 95%). The importance of a reliable CV estimate is one of the most critical steps and needs to be evaluated before sampling campaign. To achieve this, two common ways are generally proposed: i) exploiting historical data or previous studies in similar areas; ii) carrying out an exploratory analysis in test areas. For this study case, a high-quality, detailed and recent dataset was available for two study areas in Central Italy where more than 4000 *Pins nigra* trees were geo-referenced and measured in two evenly aged 50 year-old stands. The data stem from the A2 action of the SelPiBio LIFE+ project and are freely available in a public repository [10]. Among the provided data, the diameter, height and crown projection on the ground were the most basic. Indeed, no profile measurements were made. Firstly, 80 trees were randomly sampled by means of a stratified random using the DBH classes and the quartiles as strata (20 trees × 4 quartiles = 80 trees in total). The elements of this exploratory sample were located thanks to the availability of the coordinates and of the ID number marked on the stem and measured for stem taper. Afterwards, the remaining trees to be measured were measured in a second-step activity and added to this starting dataset.

In order to use Equation (1), data were firstly standardized to deal with their use in profile functions (i.e., relative diameter as a function of relative height). While all the heights were transformed in proportion of the total height of the trees, the same was done for diameters which were divided by the diameter at breast height. Then, the CV values of the measured diameters at any measurement distance from the ground (i.e., $CV_{10\%m}$, $CV_{15\%m}$, $CV_{20\%m}$, etc.) were calculated to obtain an $n$ value for each stratum using Equation (1). Then, the maximum $n$ value in the range 10%–90% was used as the reference number of samples to be representative at any section of the stem, except the top, which was not of interest and was too variable to be sampled in a cost-effective way. The final $n$ value was estimated iteratively. Indeed, the sample size $n$ conditions the determination of $t$. The starting values and degrees of freedoms (DF) were first estimated using an $n_0$ value equal to $n$-1 (with $n_0 = 80$ in this case). Then, the calculation of $t_0$ was performed to find $n_1$ samples. Afterwards, the DF were recalculated again using $_{n1}$ samples and a new $t_1$ was estimated and so on, until $n_x$ stabilized.

### 2.2. Stem Taper Functions

Stem taper functions model the relative decrease of diameter (rD) as the relative height (rH) increases. As the reference diameter, defining the relative value, possible options are the maximum value along the stem or the diameter at breast height (DBH). However, the relative decrease of the diameter should be referred to a directly measurable and easy to access parameter. For this reason, in this paper, the collected data were standardized by dividing the diameter we measured in each tree at each sampling height by the DBH. Four equations were tested, including polynomials of degrees from 1 to 3 as well as a non-parametric model. The selected models correspond to a simple linear model—Equation (2)—second and third order polynomial models—eq. (3) and (4), respectively—and a Generalized Additive Model (GAM)—eq. (5). The relative diameter of the stem measured at each sampling height ($rD_i$) was modelled as a function of the relative height of the tree ($rH_i$):

$$rD = \alpha + \beta \cdot rH + \varepsilon \tag{2}$$

$$rD = \alpha + \beta \cdot rH^2 + \gamma \cdot rH + \varepsilon \tag{3}$$

$$rD = \alpha + \beta \cdot rH^3 + \gamma \cdot rH^2 + \delta \cdot rH + \varepsilon \tag{4}$$

$$rD = s(rH) + \varepsilon \tag{5}$$

where and $\alpha$, $\beta$, $\gamma$ and $\delta$·are the coefficients to be estimated and $\varepsilon$ is the error term representing the amount of remaining unexplained deviation. Concerning GAM, the notation $s(\dots)$ denotes the use of the smoothing function for modelling.

The selection of the best performing model was then based on the underlying physical-biological process (i.e., the expected shape) of the rD–rH relationship as well as the possibility to replicate (i.e., use) the model beyond the compiled dataset and the here-described research work. In order to derive

indications on the goodness of fit, the predictions from the four mathematical models were tested by a cross-validation procedure reserving 25% randomly extracted from the database for this purpose. The mean relative absolute error (MAE) and the fraction of explained variance or r-squared ($R^2$) were used as indicators and to test whether the sampling strategy generated fair models with an error below the threshold we used in Equation (1), which was 5%. The random extraction was repeated 1000 times and single-run MAE and $R^2$ values were then averaged to an obtain unbiased estimation of goodness of fit. The best performing model was finally compared with the output from TapeR package [27] an object-oriented tool where more complex equations are implemented, such as the cubic regression models using $\beta$-Splines and mixed models, running in R statistical language [29]. The setup we used for the TapeR package was four knots positioned at 0.0, 0.12, 0.75, 1.0 relative heights and a fourth order spline function for fixed effects (cubic). Then, we used three knots positioned at 0.0, 0.1 and 1.0 and a fourth order spline function for random effects (cubic).

*2.3. Measurement Technique*

In this work, profile measurements of model trees were carried out with a non-destructive method on standing trees by use of an electronic relascope, the CRITERION RD 1000 by laser technology® (https://www.lasertech.com/Criterion-RD-1000.aspx). This optical instrument allows to measure the diameter of every section of the stem on standing trees at any visible height chosen by the user, with the horizontal distance between the operator and the tree as the parameter to adjust the optical scope. Thanks to this tool, all selected trees were fully measured, including all the social ranks, dimensions and generating an unbiased sample for stem taper equation fitting. The stem profile of all the selected tree was here fully measured with the CRITERION RD 1000, measuring the diameter every meter from the ground to the top. The instrument was combined with a TruPulse360B Laser Rangefinder form Laser Technology (https://www.lasertech.com/TruPulse-Laser-Rangefinder.aspx) using a serial cable to record distance from the target tree. The distance from the trees is a key parameter for the optical relascope in order to derive metric values from the bandwith the operator set in the instrument's scope for measurement. Overall, the average distance from target trees ranged between 15 and 20 meters, which allowed a clear view of the stem reducing distortion. The quality assessment of measurements was made by comparing the first three measurements we obtained with CRITERION RD 1000 (i.e., ground level, 1 meter and 2 meters from the ground) with the values we measured with a calliper assessing an accuracy around 98%.

## 3. Results

Based on the 80 trees measured as the pilot sample and the iterative process, the size of the final sample required to estimate the mean of the most sensitive sections (between the ground and 90% of the total height of trees) with an error within 5%, at a 95% confidence level, was estimated to be just above 207. However, the iterative process on sampling data stabilized on a value or 202 trees to be measured for an error equal to 5% (Table 1) The variability of the relative diameter increased as the relative distance from the ground increased with a maximum CV of 1.783 at the top. In the collected sample trees, the maximum height was 23.5 metres, a measure reached by only a few trees while the average height of the stand was around 18 m for both study areas. Concerning the structure of analysed stands and according to the collected data, stem diameters at central height classes, between 11.5 and 14.5 metres from the ground, were detected as the most variable. Even if, apparently in contrast with wood anatomy where the bottom of the stem is usually observed as the most variable, due to many causes such as the steep slope of the ground, the presence of basal buttresses of the stem etc., this aspect is an expected result given the structure of the stands we studied. No thinning intervention was applied since its establishment around 1950s [3,5]. This led to a stratified stand with many suppressed trees still alive and reaching heights between 11 and 14 metres. Therefore, such small diameters we measured with the optical relascope, increased the CV of the stratum and consequently, the number of trees to be

measured. However, the standardisation of the data and the calculation of the relative diameter ($rD_i$) simplified the model fitting, cleaning the database and allowing models to work properly.

**Table 1.** Coefficient of variation for each cross-section of *Pinus nigra* trees measured with CRITERION RD 1000 calculated using the 80 trees measured in the first stage sampling and derived number of trees (*n*) to be fully measured to achieve a statistical error lower than or equal to 5% according to Equation (1) and iterative self-calibrating estimation of sample size using *p* = 0.95, $\varepsilon$ = 5% and a starting at *n* = 80.

| Relative Height Class | Mean Relative Diameter | Standard Deviation | CV | Number of Samples |
|---|---|---|---|---|
| 0.00–0.05 | 1.028 | 0.023 | 0.022 | 1 |
| 0.05–0.10 | 0.988 | 0.020 | 0.021 | 1 |
| 0.10–0.15 | 0.941 | 0.035 | 0.037 | 2 |
| 0.15–0.20 | 0.890 | 0.035 | 0.039 | 2 |
| 0.20–0.25 | 0.842 | 0.040 | 0.047 | 3 |
| 0.25–0.30 | 0.798 | 0.043 | 0.055 | 5 |
| 0.30–0.35 | 0.758 | 0.047 | 0.063 | 6 |
| 0.35–0.40 | 0.716 | 0.049 | 0.068 | 7 |
| 0.40–0.45 | 0.680 | 0.051 | 0.075 | 9 |
| 0.45–0.50 | 0.635 | 0.052 | 0.082 | 10 |
| 0.50–0.55 | 0.594 | 0.058 | 0.098 | 15 |
| 0.55–0.60 | 0.552 | 0.057 | 0.104 | 17 |
| 0.60–0.65 | 0.504 | 0.064 | 0.126 | 24 |
| 0.65–0.70 | 0.456 | 0.067 | 0.146 | 33 |
| 0.70–0.75 | 0.404 | 0.075 | 0.185 | 52 |
| 0.75–0.80 | 0.345 | 0.072 | 0.209 | 67 |
| 0.80–0.85 | 0.278 | 0.077 | 0.277 | 118 |
| 0.85–0.90 | 0.194 | 0.070 | 0.363 | 207 |
| 0.90–0.95 | 0.105 | 0.056 | 0.536 | 441 |
| 0.95–1.00 | 0.017 | 0.030 | 1.783 | 4885 |
| **Iteration** | **DF** | **t** | **Number of Samples to Be Measured** | |
| 1 | 79 | 1.9905 | 206 | |
| 2 | 205 | 1.9716 | 202 | |
| 3 | 201 | 1.9718 | 202 | |

The fitting ability of the four tested models is summarised in Table 2 and graphically shown in Figure 2. Overall, very low MAE with small associated standard deviations were calculated with the cross-validation procedure. $R^2$ values were always higher than 0.95 and all the models were statistically significant at a 95% confidence level. In this scenario, only slight differences were detectable between the models. GAM was the best performing algorithm and the linear model was the worst. This was also reflected by the slightly lower $R^2$ and higher MAE. Concerning the remaining GAM and third-order polynomial models, only the last one was selected for a further comparison with TapeR prediction. This was due to the complexity of GAM to run on external data and outside the modelling framework (i.e., the R environment in this case), even if characterised by a better fitting.

**Table 2.** Cross validation results.

| Model | Mean Relative Absolute Error | Explained Variance |
|---|---|---|
| Linear | $0.03273$ ($\pm 0.96 \times 10^{-3}$) | 96.1% ($\pm 0.60 \times 10^{-3}$) |
| Second order polynomial | $0.03195$ ($\pm 0.85 \times 10^{-3}$) | 97.1% ($\pm 0.55 \times 10^{-3}$) |
| Third order polynomial | $0.01362$ ($\pm 0.83 \times 10^{-3}$) | 97.4% ($\pm 0.49 \times 10^{-3}$) |
| GAM | $0.01254$ ($\pm 0.84 \times 10^{-3}$) | 97.5% ($\pm 0.50 \times 10^{-3}$) |

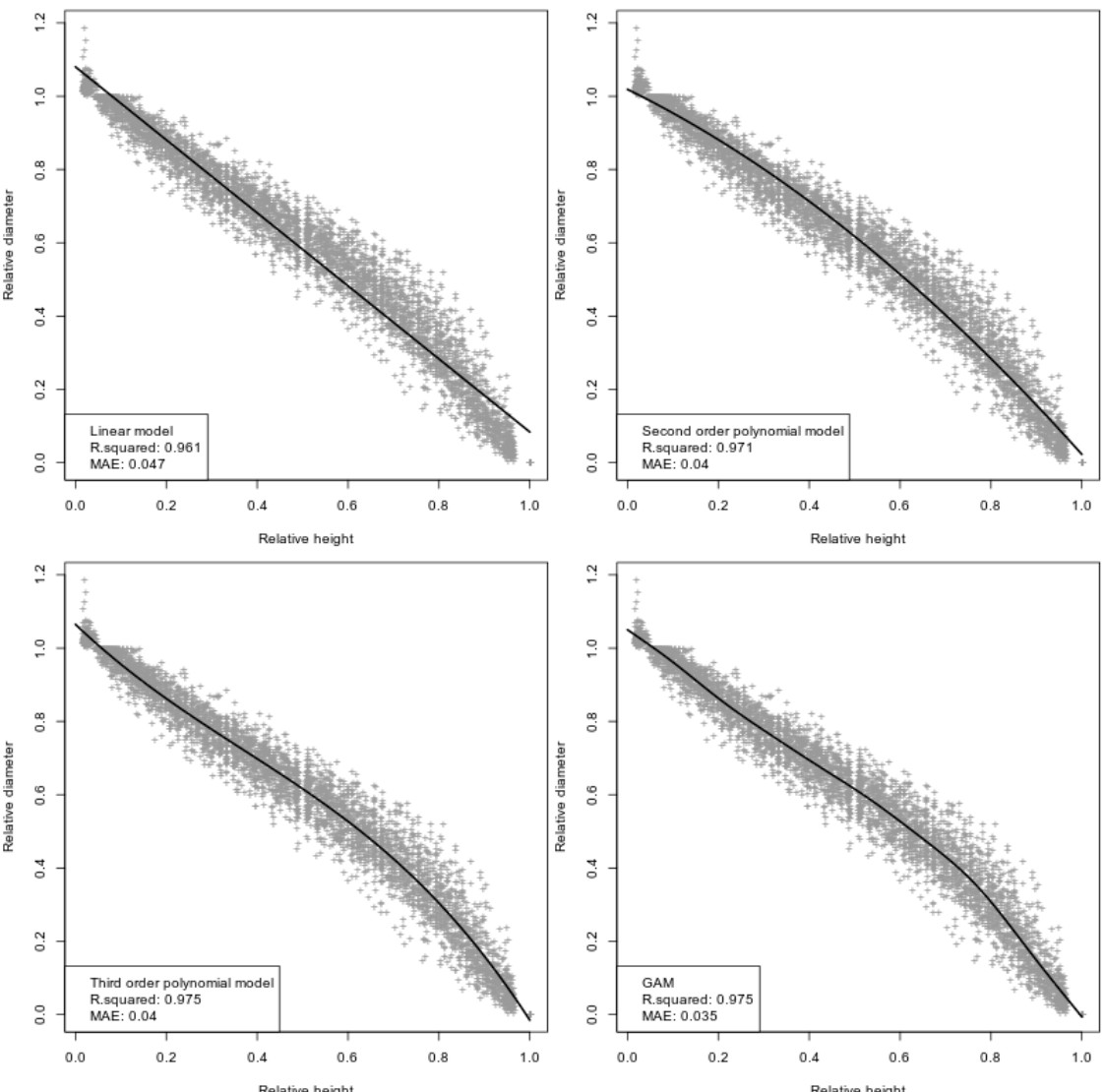

**Figure 2.** Fitted models on the whole database using the four models. Confidence intervals are not displayed because they were too small and not possible to be observed in a small image.

MAE values estimated by cross-validation are actually much lower than the maximum tolerated 5% error value used to determine sample size. MAE values appear quite constant for all relative heights (Figure 3). The sample size estimation procedure could possibly be improved in order to reduce oversampling in case of greater budget limitations.

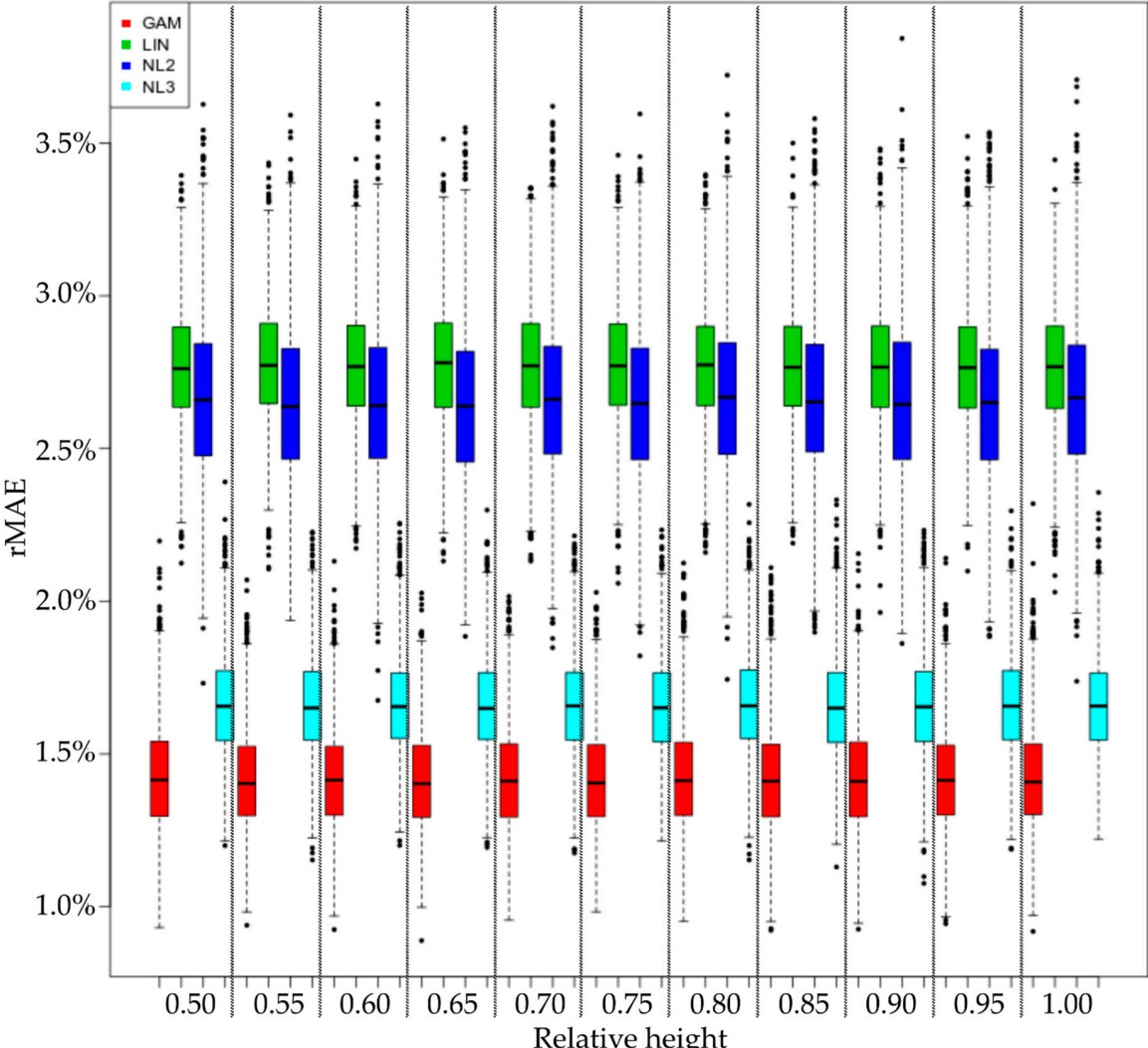

**Figure 3.** MAE of tested models at different relative height along the stem. The boxplots are colored according to the legend in the figure.

Finally, the third-order polynomial function was selected as the most simple and effective model. Calibrating it using the complete data set, the final coefficients were estimated and are reported in Table 3. The inflection was detected around rH = 0.55, demonstrating a sort of change of the relationship between rD and rH almost in the middle of the range of the data.

The comparison between the third-order polynomial model estimates and TapeR shows that all the predictions of the simple parametric model were included within the 95% confidence interval provided by TapeR. When plotting the estimates from the two techniques for a simulated tree, no significant differences were apparently found. This comparison is graphically shown in Figure 4 in which the two models were used to simulate the profiles of two hypothetical groups of *Pinus nigra* stems, one with a fixed height and variable DBH, the other with a fixed DBH and variable height. Indeed, the estimates of the two methods were linearly correlated with highly statistically significant parameters ($p < 0.0001$, cor = 0.99).

**Table 3.** Estimated coefficients for the third order polynomial function.

| Coefficient | Estimate | Standard Error | t value | Pr(>|t|) | |
|:---:|:---:|:---:|:---:|:---:|:---:|
| $\alpha$ | 0.590 | 0.00072 | 817.57 | $< 2.2 \times 10^{-16}$ | *** |
| $\beta$ | −1.265 | 0.03827 | −25.92 | $< 2.2 \times 10^{-16}$ | *** |
| $\gamma$ | −2.075 | 0.04882 | −42.50 | $< 2.2 \times 10^{-16}$ | *** |
| $\delta$ | −20.44 | 0.02213 | −418.71 | $< 2.2 \times 10^{-16}$ | *** |
| *p*-value: $< 2.2 \times 10^{-16}$ - *R*-squared: 0.9749 - Residuals Standard error: 0.04882 | | | | | |

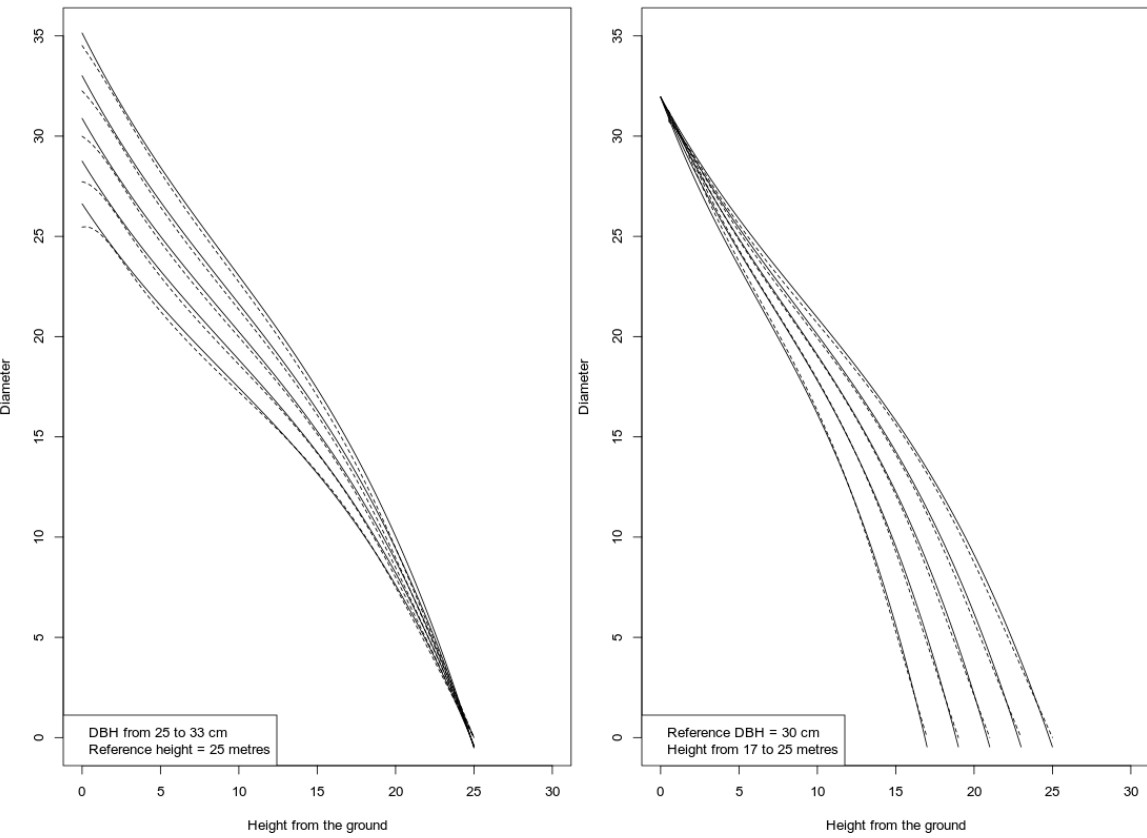

**Figure 4.** Stem profile generated using the third order polynomial function (continuous line) and TapeR package (dashed line) with two different hypothetical groups of *Pinus nigra* trees with same height but different DBH (left) and the opposite study case (i.e., same DBH but different height).

## 4. Discussion

The use of adequate models both for prediction and data collection (i.e., balancing the sample size) deeply affected the results of this experiment. Stem taper functions are basic tools in forest management practices to derive information on the value of the timber yield and the statistical evaluation of the sample size offers a means for a rational approach to sample size optimization. The non-destructive two-stages sampling strategy presented here stressed the importance of the data as the focal point for modelling to derive reliable and accurate stem taper functions presents high data requirements. While the effective MAE was much lower than selected relative error (5%) we used, in Equation (1), reliable models were built with all the four tested equations. Then, concerning the sample balancing, Equation (1) proved to be too general and not reliable. However, this result is in line with previous studies in forest monitoring where the same issue was found [28]. Model selection has often been acknowledged as one of the most critical steps by modellers in many research fields, correctly achievable through a statistical comparison [30–32]. Several studied also focused on data quality when dealing with

climate [33,34], forest mensuration [35,36] and modelling activities in general [37–39]. While data quality was detected here once more as the focal point for cost-effective research, the same cannot be said for modelling tools. A small difference within the group of parametric/non-parametric models was found here but a wide discrepancy between them and the mixed-modelling TapeR package was determinant for the model selection. In our case, while the linear model and the second-order polynomial function were simply inadequate to represent the sigmoidal shape of the point cloud we obtained, all the equations were not able to handle the autocorrelated structure of the data we measured. Conversely, the mixed-model approach can handle the hierarchical structure of the data where the assumption of the independency between sampling units is violated. The profile measurements of trees are a typical example in which the measurement made at height *H+h* is strongly correlated to the data collected at height *H* [27].

The economic sustainability of forest harvesting in artificial *Pins nigra* spp. as well as in the case of many other forest tree species has often been acknowledged as one of the main shortcomings for the application of successful forest management [5,40]. Many *Pinus nigra* stands originally planted in the 20th century in Italy (but also in many other EU countries) are quite like those we studied and are still abandoned to natural dynamics even if characterised by an increasing biodiversity level and ecological success [21,22]. The use of the provided model might support renewed interest around this species, allowing forest enterprises and stakeholders in general to plan forest harvesting according to the expected timber potentially achievable from a specific forest stand. This could also be positively seen by research in order to test the provided equation and implement novel modelling tools in addition to stem taper equations. The use of a small spatial extent for sampling might be a potential shortcoming of the provided models where "just" 202 trees were measured from two stands. This issue might not allow our model to be applied for estimates in other regions where different growth trends might occur.

Even if further research efforts are necessary to test and validate the provided model, the genesis of analysed stands, i.e., artificial stands with seeds coming from different parts of Italy [8,11] and, consequently, the possible mixture of considered genotypes, could be seen as an additional positive trait of this study. In any case, the low variability of the dataset we compiled, and the high predictive power of TapeR package needs to be confirmed by additional study cases. Even if object-oriented tools and portals were packed by modellers or informatic engineers with more sophisticated modelling methods such as Neural Networks or Random Forests algorithm, the use of mixed-effects models like TapeR seems to be compulsory for unbiased estimates. The idea of running simple equations outside the framework of a programming language such as R, Phyton or MATLAB should be, in our opinion, discouraged in favour of web-tools and cloud computing systems able to exploit the full functionality of statistical environments, such as R, even by common users. The development of taper functions cannot be limited to simple equations on sheets. Forest planning and management activities require timber volume estimations with adequate accuracy and taper functions are developed in order to improve timber volume estimation capacity and particularly the accuracy of timber value estimation, as the volume is divided into assortments [5,41]. Growth and yield models are used to provide longer-term scenario evaluations and to develop planning decisions [16,42,43].

In other ordinary processes like the detailed planning of harvesting operations, the requirement is to estimate timber volume, possibly by assortment, for given stands [12]. In this case, more general-purpose software environments are used, typically spreadsheets. Timber volume estimation functions are generally sufficiently simple. Ordinary operators can implement the functions required in the spreadsheet computations and a relatively simple model representing a taper function offers the opportunity for an ordinary user of spreadsheets to implement it and perform some testing and get acquainted with the tool. However, an effective use of the function as a tool for optimizing the subdivision of the total volume in the desired assortments is not obtainable with only basic capacity in the use of spreadsheets. Taper estimation implies, in the optimization process, a pair of functions: one estimating the diameter of the cross section at any given height, and the second coherent with the first, estimating up to which height the diameter of the cross section is greater than a given threshold [19,27].

Looking shortly ahead of the current situation, basic coding competencies and skills will become unavoidable as cloud-based environments extend and forest operators will need to connect databases and processing modules to perform the tally, store temporary and final measurements and observations, process the data and produce the required reports.

## 5. Conclusions

The paper title includes the question "is a more complex model required?". The model finally selected as a taper function for the studied *Pinus nigra* stems is quite simple; it is a third-degree polynomial estimating the relative diameter for any given relative height along the stem. A competitive solution is the output that R-package TapeR can provide. While implementing our solution in a spreadsheet or in any programming framework is straightforward, the competitive solution can be conveniently used to develop estimation procedures within the R environment, but it is quite complicated to transfer to other environments.

Since the use of stem taper equations is still rare in the forestry sector, despite their potential as basic forest management tools, the complexity of the tool has a relevant impact on the possibility for their use to spread. To this end, the simple solution developed offers several opportunities for foresters that are not specialised in coding to get in direct touch with the tool.

An option for more complex models to be made accessible by non-specialists could be providing Internet-based solutions. TapeR functions and the parameter set that can be estimated with the package can be incapsulated in the shiny web app (https://shiny.rstudio.com/) which can be a powerful environment where also non-statistician users might be able to generate stem taper profiles using more complex $\beta$-Splines and mixed models.

**Author Contributions:** Conceptualization, M.M. and P.C.; methodology, M.M., R.S. and P.C.; software, M.M. and R.S.; validation, M.M., R.S. and G.R.; formal analysis, M.M. and R.S.; investigation, G.R., P.C. and M.M.; data curation, M.M., R.S. and G.R.; writing—original draft preparation, M.M. and G.R.; writing—review and editing, M.M., R.S. and P.C.; supervision, R.S. and P.C.; project administration, P.C.; funding acquisition, P.C. All authors have read and agreed to the published version of the manuscript.

**Funding:** The research was funded by the SelPiBio LIFE + project "*Innovative silvicultural treatments to enhance soil biodiversity in artificial black pine stands*" (LIFE13 BIO/IT/000282), https://www.selpibio.eu/.

**Acknowledgments:** A special thanks to Valter Cresti and Luca Marchino from CREA—Research centre for Forestry and Wood for their help in field activities.

**Conflicts of Interest:** The authors declare no conflict of interest.

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
