# Peer review of "Taper Function for Pinus nigra in Central Italy: Is a More Complex Computational System Required?"

_forests, doi:10.3390/f11040405_

Round 1
Reviewer 1 Report
Two research plots seems to be not apropriate if there is some influence of local conditions for the results, therefore this should be explain in Methods. In Conclusion would be better to repeat the main results.
Author Response
The potential impact of just two study areas was already included in the Discussion section. This because we believe this issue should not be seen as a potential lack of our study/model but more a problem in case of using our models (i.e. coefficients) outside the study area. That’s why we discussed it there. Then the conclusion section was restructure as also suggested by Rev#2

Reviewer 2 Report
General comments
In the manuscript the Authors present several simple models for constructing a taper function for two black pine stands in Italy. The manuscript reports on the results from a study of local character. The novelty of the study is very low. The most important result, in my view, is the set of parameters for 3rd order polynomial taper function obtained by simple non-linear fitting. In addition to low novelty, the presentation of the manuscript does not rise to the standards I would expect for the publication in high IF journal. Some key information is missing (details below). Results partly focus important, but rudimentary issues (sample size) – it could be reported in one paragraph without the need of detailed elaboration. Discussion is also lengthy, mostly general and not sufficiently focused on the results. The conclusions are not conclusions at all.
Specific objections:
Formula 1 is not correct.
It is unclear how many trees were measured in in total (80, 207?) and how many in each stand? Did you actually first measured 80 trees, and only afterwards measured remaining 127?
The Authors used Criterion 1000 RD instrument for non-destructive sampling. They did not mention if they used distance meter, and which kind of they used.
The accuracy of their measurement with the relascope was not tested nor reported. Visibility issues with the higher parts of the stem in medium dense to dense stands would occur when measuring from optimal distance (in this case 15-20 m). How it was dealt with?
The use of the mentioned TapeR package in R statistical software is not described, namely: Which settings were used? Which taper function (“type”) was selected? What were the required input and outputs?
The map shown in Figure 1 is already published in [28]. The stands where the measurements were made are not designated on the map.
Figure 5 – Figure description in the caption is not detailed enough (Which model is shown? For how many trees? – also I can only guess that the red line represents “TapeR”, as it is not written).
Author Response
Authors: We are sorry the reviewer evaluates our paper as “low novelty” and below “the standards I would expect for the publication in high IF journal”. In our opinion the paper was scientifically robust, interesting for a wide audience and able to stimulate discussion around a non-conventional topic. Stem taper functions are quite uncommon in literature and non-destructive sampling methods are just arising nowadays by means of optical relascopes and laser scanning techniques.
We worked to improve the text adding the key information he/she asked (details below) also addressing the issue for the Results and Discussion sections. Then the conclusion section was restructure as also suggested by Rev#1
Specific objections:
Formula 1 is not correct.
Authors: a typo error. The formula has been fixed
It is unclear how many trees were measured in in total (80, 207?) and how many in each stand? Did you actually first measured 80 trees, and only afterwards measured remaining 127?
Authors: The text was probably confusing with also a typo error concerning the total number of trees we measured which was 2002 and not 207. Actually, what we did was to firstly measure 80 trees; then with the derived CV we used the iterative process to balance the dataset enrichment, which was composed, in the end, by 122 trees which were measured in a second step. We modified the text to make it clearer
The Authors used Criterion 1000 RD instrument for non-destructive sampling. They did not mention if they used distance meter, and which kind of they used.
Authors: The distance is a key parameter for measuring and you are right that an additional explanation was necessary. We added this in the text enlarging section 2.3
The accuracy of their measurement with the relascope was not tested nor reported. Visibility issues with the higher parts of the stem in medium dense to dense stands would occur when measuring from optimal distance (in this case 15-20 m). How it was dealt with?
Authors: The issue the reviewer addressed is important and we apologise for the shortcoming. We added this in the text enlarging section 2.3
The use of the mentioned TapeR package in R statistical software is not described, namely: Which settings were used? Which taper function (“type”) was selected? What were the required input and outputs?
Authors: a wider explanation was added in the text allowing the user to replicate our study. Briefly we used four knots for fixed effects positioned at 0.0, 0.12, 0.75, 1.0 relative heights, a fourth order spline function for fixed effects (cubic), three knots for random effects positioned at 0.0, 0.1 and 1.0 and a fourth order spline function for random effects (cubic).
The map shown in Figure 1 is already published in [28]. The stands where the measurements were made are not designated on the map.
Authors: The reviewer’s question doesn’t seem correct to us. He/She is probably confusing our figure (which is novel and not published elsewhere) with Figure 1 in reference [28] which is Marchi, M.; Ferrara, C.; Bertini, G.; Fares, S.; Salvati, L. A sampling design strategy to reduce survey costs in forest monitoring. Ecol. Indic. 2017, 81, 182–191 and where the spatial distribution of ICP-Forests plots (Level I and Level II) in Italy is shown, not the INFC2005 survey plots. Conversely, he is right writing that no figures are provided with the stands where the measurements were made. However, we prefer not to include this information in our paper for redundancy. While the dataset is already summarised in section 2.1 the reference [10] connects this paper to the dataset paper we published in 2017, the SelPiBio LIFE dataset. Then it is true that the trees we really measured (a small subset of the dataset) are not shown but a map with two squares with only 100 points in 15 hectares area each wouldn’t be so informative.
Figure 5 – Figure description in the caption is not detailed enough (Which model is shown? For how many trees? – also I can only guess that the red line represents “TapeR”, as it is not written).
Authors: The figure was probably confusing for the reader and we decided to remove it

Round 2
Reviewer 1 Report
Thank you for the explanation and for the text revision. This seems to me more comprehensible. It is rather a guide on how to proceed in the evaluation of stem tapers, so I believe that local validity is not as fundamental as it seemed to me when revising the first version.
Author Response
Authors: Thanks for your comment. We are glad this new version meets your ideas
Reviewer 2 Report
Authors considerably improved the materials and methods section. The description of the research method is now clear and complete, thus allowing the replication of the study by others elsewhere. Authors also made some additional edits to the text making the manuscript easier to read.
Page 3.
TLS - Terrestrial Laser Scanning? Define TLS when first used in the text.
Page 4.
"(i.e. relative diameter ad a function of relative height)" - typo "ad" -> "as"
Author Response
Authors: Thank you and we are happy that our amendments have satisfied you. The two small issues you suggested below have been corrected in the text